# Impact of excessive social media use on adolescent depression and its consequences in France: An individual-based microsimulation model

Nicolas Hoertel [1,2,3*], Mark Olfson[4], Carlos Blanco[5], Margot Biscond[6], Frédéric Limosin[1,2,3], Marina Sánchez-Rico[1], Martin Blachier[6], Henri Leleu[7]

1 AP-HP.Centre, DMU Psychiatrie et Addictologie, Corentin Celton Hospital, Issy-les-Moulineaux, France, 2 Université Paris Cité, Faculté de Santé, UFR de Médecine, Paris, France, 3 INSERM UMR_1266, Institut de Psychiatrie et Neuroscience de Paris, Paris, France, 4 New York State Psychiatric Institute, Columbia University Irving Medical Center, New York, New York, United States of America, 5 Division of Epidemiology, Services, and Prevention Research, National Institute on Drug Abuse, Bethesda, Maryland, United States of America, 6 Public Health Expertise, Paris, France, 7 Public Health Expertise, Paris, France

* nico.hoertel@yahoo.fr, nicolas.hoertel@aphp.fr

## Abstract

### Background

Social media (SM) platforms have become increasingly prevalent in adolescents' lives, and concerns have arisen regarding their potential contribution to depression. This study examined whether excessive SM use contributes to rising adolescent depression rates and evaluated potential mitigation strategies.

### Methods and findings

We developed an individual-based microsimulation model of 18.6 million French adolescents born 1990–2012, tracking depression outcomes from 2000–2022 (analyses conducted August 2024-July 2025). The model incorporated 95 parameters, including demographics, SM use patterns, and established depression risk factors (childhood adversities, chronic physical conditions, physical inactivity, obesity, substance use). The main outcome was cumulative depression cases, and secondary outcomes were suicide deaths, health-adjusted life expectancy (HALE) loss, and associated costs. The model was well-calibrated and validated adequately against US-specific data. It showed that excessive SM use likely played an important role in the recent increase in rates of adolescent depression. Among French adolescents, simulations indicated that excessive SM use was associated with an additional cumulative lifetime 590,000 depression cases (95%CI [400,000, 760,000]), 799 suicide deaths (95%CI [547, 1,028]), 137,000 (95%CI [94,000, 176,000]) HALE loss years, and 3.94 (95%CI [2.70, 5.07]) billion euros, compared to scenarios without SM platforms. Key limitations are

**Data availability statement:** Population demographics, distributions of chronic physical conditions, obesity and overweight, childhood adversities, physical inactivity, substance use in France are obtained from publicly available data of French National Statistical Institute (INSEE) [1]; [https://www.insee.fr/fr/statistiques/fichier/8327319/ip2033.xlsx], Santé Publique France [2,3,4,5]; [https://doi.org/10.1016/j.eujim.2021.101308; https://doi.org/10.1016/j.respe.2022.01.110; https://www.santepublique-france.fr/maladies-et-traumatismes/sante-men-tale/depression-et-anxiete/documents/article/prevalence-des-episodes-depressifs-en-france-chez-les-18-85-ans-resultats-du-barometre-sante-2021; http://beh.santepubliquefrance.fr/beh/2020/15/2020_15_1.html], Observatoire Français des Drogues et des Tendances Addictive (OFDT) [6–9]; [https://www.ofdt.fr/publication/2023/evolution-de-l-usage-regulier-d-alcool-17-ans-depuis-2000-1758; https://www.ofdt.fr/publication/2023/usage-quotidien-de-tabac-par-sexe-17-ans-evolution-depuis-2000-1761; https://www.ofdt.fr/publication/2023/usage-au-cours-de-la-vie-de-cannabis-parmi-les-17-ans-evo-lution-depuis-1993-1762; https://www.ofdt.fr/publication/2025/usage-au-cours-de-la-vie-de-substances-psychoactives-hors-alcool-tabac-cannabis], Ministère des Solidarités et de la Santé [10,11]; [https://drees.solidarites-sante.gouv.fr/publications/etudes-et-resultats/en-2017-des-adolescents-plutot-en-meil-leure-sante-physique-mais, https://drees.solidarites-sante.gouv.fr/sites/default/files/er283.pdf], l'École des Hautes Études en Santé Publique (EHESP) [12]; [https://doi.org/10.1016/j.jad.2022.11.078], and the French portion of the World Mental Health International College Student survey. Data on social media use are available from the publicly available data of Mediamétrie [13]; [https://www.mediametrie.fr/]. Data on prevalence rates of adolescent depression in France and in the United States are publicly available from l'Enquête santé européenne (EHIS) and l'Enquête nationale Épidémiologie et Conditions de vie (EpiCov) (Inserm-DREES), a systematic literature review [14]; [https://doi.org/10.1002/da.22777] and the National Survey on Drug Use and Health [15]; [https://doi.org/10.1016/j.jadohealth.2021.08.026], respectively. Data

that microsimulation modeling cannot establish causality from observational data and the reliance on duration-based exposure measures without capturing content type or engagement quality.

## Conclusions

In this study, we estimated that limiting SM use to 1 h per day for all adolescents, replacing 30 min of SM use with 30 min of physical activity, or stopping its use for adolescents most at-risk for depression, would be associated with a reduction in cumulative lifetime prevalence of depression by 14.7%, 12.9%, and 12.0%, respectively, and diminished associated costs. Targeted SM interventions could potentially reduce adolescent depression burden, though real-world implementation and effectiveness require validation.

---

## Author summary

### Why was this study done?

- Depression rates among teenagers have risen dramatically over the past decade, but the causes remain unclear.

- Social media (SM) use became widespread during this same period, raising concerns about its potential role in depression.

- Previous studies showed conflicting results about whether SM is associated with depression.

### What did the researchers do and find?

- Development of a microsimulation model comprising 95 parameters tracking depression outcomes in 18.6 million French adolescents born between 1990 and 2012.

- The model aimed to test whether excessive SM use contributes to rising adolescent depression rates and to evaluate mitigation strategies.

- The model projected that excessive SM use was associated with 590,000 additional lifetime depression cases and 799 suicide deaths among French adolescents.

- Simulations suggested that limiting SM to 1 h daily could reduce depression rates by 14.7%.

### What do these findings mean?

- Excessive SM use may substantially contribute to rising adolescent depression rates.

inconsistencies were resolved by prioritizing recent national surveys, and missing values were handled through linear interpolation or conservative assumptions within our ±20% uncertainty range. The source code of the model has been deposited in a recognized public source code repository (GitHub, https://github.com/henrileleu/MicrosimDepressionSM) and is permanently archived on Zenodo with a DOI: https://doi.org/10.5281/zenodo.16992932.

**Funding:** The author(s) received no specific funding for this work.

**Competing interests:** The authors have declared that no competing interests exist.

**Abbreviations:** CI, confidence interval; EHESP, l'École des Hautes Études en Santé Publique; HALE, health-adjusted life expectancy; MAE, Mean Absolute Error; OR, odds ratio; RMSE, Root Mean Square Error; SM, social media; TRIPOD-AI, Transparent Reporting of a Multivariable Prediction Model for Individual Prognosis or Diagnosis.

- These microsimulation findings suggest targeted SM interventions could potentially reduce adolescent depression burden.

- Real-world implementation and effectiveness of these interventions require validation.

- Limitations include inability to determine causality and potential under-reporting of SM use.

## Introduction

Depression prevalence rates in adolescents and young adults have risen sharply in the past decade, especially in females [16–19], which represents a growing major public health concern [20]. Depression is a leading contributor to the global burden of disease [21] as it is associated with large personal, societal, and economic burden, especially when it occurs during a key life transition period marked by rapid emotional, cognitive, and social development [20,22]. While risk factors for depression in this population are well-established, including family history of depression, exposure to social stressors (e.g., childhood maltreatment, bullying, stressful life events), substance use, and belonging to certain subgroups (e.g., having a chronic physical health problem or being a sexual minority) [20,23–25], the recent increase in the prevalence of depressive symptoms among adolescents [26,27] remains poorly understood [20]. A better understanding of its causes is necessary to address this public health priority [28] and help guide preventive and early intervention strategies.

In recent years, social media (SM) platforms have become increasingly prevalent in the lives of children and adolescents [29]. According to the Pew Research survey, 89% of US teens report their SM use to range from several times a day to constant [29]. While these platforms may offer some benefits in terms of self-expression and social connections, concerns have arisen regarding their potential deleterious effects on the mental health of certain adolescents and young adults [30–35], and especially on their risk to induce depressive symptoms in some young people. For instance, Twenge and colleagues [30] found that adolescents who spent more time on new digital media (including SM and electronic devices such as smartphones) were significantly more likely to report lower psychological well-being, including less curiosity, lower self-control, more distractibility, more difficulty making friends, less emotional stability, being more difficult to care for, and an inability to finish tasks. By contrast, those who spent more time on non-screen activities (such as in-person social interactions, sports or exercise, homework, print media, and attending religious services) were less likely to report these outcomes. Similarly, Sampasa-Kanyinga and colleagues [31] found that daily social networking site use for more than 2 h was associated with poor self-rating of mental health, high levels of psychological distress, and suicidal ideation in a representative sample of 753 middle and high school children in Ottawa, Canada. A recent systematic literature review and meta-analysis [32], including 21 cross-sectional studies and 5 longitudinal studies ($N = 55{,}340$ adolescent participants) showed a small, but

significant association (odds ratio (OR)=1.13; 95% confidence interval [CI] [1.09, 1.17]; $p < 0.001$) between time spent on SM and depression among adolescents, especially teenage girls, in a dose–response manner. Experimental evidence also suggests that curbing SM use may bolster mental health well-being among adolescents [33–35]. Finally, recent evidence suggests the relationship between SM use and adolescent mental health could be more nuanced than simple dose-response effects. Factors such as the type of engagement (active posting versus passive scrolling), exposure to cyberbullying, the nature of online interactions (supportive versus hostile), and individual differences in vulnerability may all moderate this relationship [36,37].

Therefore, the role of SM in the increase of adolescent depression rates remains controversial, as it could either be a consequence or causally unrelated. Prior studies, mostly relying on cross-sectional data [32], have reported a mix of small positive, negative, and null associations [32], which could be explained by recall and unmeasured confounding biases. Indeed, other factors such as physical inactivity [38], obesity [39,40], exposure to cyberbullying, and engagement to environments in which SM are used [32,41] could confound this association. Since randomized clinical studies of SM use are logistically difficult to implement and conclusions from longitudinal studies may be compromised due to the wide range of social, psychological, and biological confounders [20,42], establishing a potential causal relationship [43] between SM use and adolescent depression is challenging.

In this report, we present results from an individual-based microsimulation model [44–48] of the impact of SM use on the risk of depressive symptoms among adolescents and its associated medical and societal costs in France. We projected the potential impact of limiting SM use for all adolescents or for those most at-risk for depression, and of replacing part of the SM time with physical activity. Due to the multifactorial nature of adolescent depression, the analysis was based on a calibration that accounts for several important known social and clinical contributing factors. Depression in this report refers to depressive symptoms and not major depressive episode, as it is the case with most previously published studies focusing on this potential association [32]. Advantages of the use of microsimulation model compared to the traditional macro-level population projection models include its flexibility and adaptability in a context of multiple possible confounding factors, such as in adolescent depression, and the possibility to evaluate diverse causation hypotheses against real-world data and to compare different scenarios to help inform researchers and policy makers [49]. In this study, we hypothesized that excessive SM use substantially contributes to rising adolescent depression rates, and that targeted SM interventions could potentially reduce adolescent depression burden.

## Methods

Individual-based microsimulation is a computational approach that models heterogeneous populations by simulating each person's life trajectory separately, incorporating their unique characteristics and risk factor profiles [44–48]. Unlike cohort-based models, this approach captures individual-level heterogeneity in risk factors, their accumulation over time, and differential responses to interventions. This methodology is particularly suited for evaluating complex policy interventions where effects vary across population subgroups. We chose microsimulation over agent-based modeling because our research question focused on how individual-level risk factors (including SM exposure duration) accumulate over time to influence individual depression outcomes, rather than examining how behaviors or mental health states spread through social networks or how peer interactions influence individual outcomes. Following previously described methods [44–48], we built an individual-based microsimulation model to test the hypothesis that SM use increases the risk of depression among adolescents. This model comprised 95 parameters grouped into three modules: (1) demographic characteristics and mortality (4 parameters), (2) SM use (7 parameters), and (3) risk factors for adolescent depression (84 parameters). Of these 95 parameters, 93 were derived directly from published literature, one was based on expert assumption, while only one parameter—the individual baseline risk—required calibration. Detailed values for all parameters and the references used to estimate their values can be found in S1 Table.

## Demographic characteristics

The model simulated birth cohorts ranging from the 1980s to the 2010s and reproduced sex and age structures of the French general population [50]. Each simulated individual was attributed a probability of death based on contemporary mortality tables for France [51].

## Social media use

SM platforms, comprising Facebook, Twitter, WhatsApp, Snapchat, TikTok, and Instagram, were included in the model to closely mirror real-world usage [52]. Each platform was attributed a probability of use for each individual aged 10–17 years, influenced by sex, age, and the year of introduction of each social medium. The adoption curves for each SM platform were modeled using the Bass diffusion framework applied to actual historical uptake data from Mediamétrie [52], with platform-specific timing based on market entry years and typical adolescent saturation dynamics. A typical adoption curve [53] for use of SM was modeled according to actual utilization data [52], starting with Facebook's introduction in 2004. The shape of the curve is described in supplementary material (S1 and S2 Figs). Daily usage durations per platform were estimated based on specific data from France [52] and were extrapolated using a gamma distribution, which appropriately captures the right-skewed nature of SM usage where most users have moderate usage while some have very high usage [54], assuming a coefficient of variation (SD/mean) of 20%. The gamma distribution parameters were applied uniformly across platforms as a simplifying assumption, even if platform-specific distributions may exist.

Consistency in individual SM use over time was ensured using a single probability value for each individual across time for each platform (S2 Table). Additionally, we assumed a tendency toward general SM consumption for individuals using a SM platform, by adding a correlation coefficient of 0.10 across SM use platform use. This modest correlation coefficient was based on observed co-use patterns in adolescent populations [52] and was tested in sensitivity analyses, showing minimal impact on overall model outcomes. The model accounted for total time spent and frequency of SM use [55].

## Main contributive factors to adolescent depressive symptoms

Because adolescent depressive symptoms results from a complex interaction of social, psychological, and biological factors [42], we considered that this risk, beyond the putative SM use factor, was influenced by sex and age [56], parental history of psychiatric disorders [57], exposure to childhood adversities [58–61], physical inactivity [62], obesity and overweight [39,63,64], chronic physical conditions [65], and substance use (i.e., alcohol, tobacco, and cannabis use) [66]. These risk factors were implemented as OR for the risk of depression when the risk factor was present in the simulated individual.

The presence of risk factors was stochastically assigned to simulated individuals accounting for age, year, and sex, based on their observed distribution in the French population. Specifically, for childhood adversities, including bullying, we used data from the French portion of the World Mental Health International College Student survey [12] and from l'École des Hautes Études en Santé Publique (EHESP) [67], respectively. Estimates of physical activity among adolescents were obtained from national school health surveys [68,69], stratified by sex, year, and two age categories [10–14 years and 15–17 years]. It was assumed that physical activity has remained constant since [69] but decreased by 42% during the COVID-19 pandemic [2]. For obesity and overweight, we used data from national school health surveys, stratified by sex [10,11] and year. Estimates of chronic physical conditions were based on data reported by the National Health Insurance [3], stratified by sex and year. For substance use, we used data from the HBSC-EnCLASS study [67], stratified by sex and year. Finally, to account for the increase of the reported cases of adolescent depression observed in France from 2020 to mid-2021, we included for 2020 [4] a specific distinct risk factor to reflect the potential impact of school closure and confinement related to the COVID-19 pandemic on adolescent depression.

Therefore, we considered the risk of adolescent depression at any given age would follow the following formula:

$$Risk(t) \sim \exp(\sum \ln(OR_i)A_{i,t} + \ln(OR_{SM})\, SM_t + I)$$

(1)

where $A_{i,t}$ are indicator variables for presence at age $t$ of known risk factors for depression (i.e., sex, parental history of psychiatric disorders, childhood adversities, physical inactivity defined as less than 60 min of activity per day, obesity and overweight, chronic physical conditions, substance use and confinement/school closure during the COVID-19 pandemic), $SM_t$ an indicator for SM use at age $t$, $OR_i$ and $OR_{SM}$ the ORs identified in the literature for risk of depression associated with the presence of each risk factor and SM use respectively, and $I$, the individual baseline risk that reflects residual risk unexplained by all other factors.

Within the model, the onset of depressive symptoms was stochastically assessed annually for each individual aged 10–17 based on the risk estimated from equation (1), accounting for variations in individual risk factors and exposure to SM. We assumed that the baseline residual risk factor, parental history of psychiatric disorders and exposure to childhood adversities were constant between ages 10 and 17.

## Outcomes

The main outcome was the lifetime cumulative number of cases of depression among adolescents between 2000 and 2022.

We also projected the lifetime medical and societal costs linked to adolescent depressive symptoms and its care into adulthood, including costs related to depression recurrence, productivity loss, health-adjusted life expectancy (HALE) loss, and suicide deaths and suicide attempts (Tables 1 and 2). Based on a systematic review [14], we considered that adolescents with depression had 2.78 (95%CI [1.97, 3.93]) times increased odds of depression in adulthood. Assuming the prevalence of adult depression of 8.3% [70] based on the 2021 National Survey on Drug Use and Health (NSDUH), this would yield an expected adult prevalence of about 23% (2.78 times 8.3%) among those who had depressive symptoms as adolescents.

For direct medical costs, average costs of care for depressive symptoms were assumed to be similar to depression and were based on the French national health insurance estimates [16]. For loss of productivity, assumptions from the

**Table 1. Summary of key model parameters and uncertainty ranges.**

| Parameters | Average | 95%CI | | Parameter role[a] | Parameter function[b] | Evidence source[c] | Reference number |
|---|---|---|---|---|---|---|---|
| **Calibrated parameter** | | | | | | | |
| Baseline risk of depression | 1.8000 | 1.8000 | 1.8000 | Calibrated | Calibration | Model fitting | – |
| **Social media use associations** | | | | | | | |
| Using social media 30–60 min per day | 0.0513 | −0.3507 | 0.4620 | Input | Risk association | Empirical | [55] |
| Using social media 60–120 min per day | −0.2776 | −0.6881 | 0.1278 | Input | Risk association | Empirical | [55] |
| **Risk factors** | | | | | | | |
| Physical activity <60 min per day | 0.4780 | 0.1278 | 0.8210 | Input | Risk association | Empirical | [38] |
| Bullying | −0.4383 | −0.5710 | −0.3001 | Input | Risk association | Empirical | |
| Physical abuse | 0.1278 | −0.0100 | 0.2614 | Input | Risk association | Empirical | |
| Male | 0.2357 | −0.0392 | 0.4943 | Input | Risk association | Empirical | [55] |
| COVID19 | −1.4816 | −1.6864 | −1.2809 | Input | Risk association | Empirical | [4] |

[a]Input (external sources) vs. Calibrated (model-fitted).

[b]Risk association (odds/hazard ratios), Calibration (fitted parameters).

[c]Empirical (primary studies), Model fitting (calibration-derived).

**Table 2. Summary of projected effects of interventions applied between 2000 and 2022 on the cumulative lifetime prevalence of depression and its associated medical and social costs among the adolescents born between 1990 and 2012 in France (*N* = 18.6 millions), as compared to the current situation (observed trends over the same 2000–2022 period).**

| | Compared to no social media usage | Compared to the current situation | | |
|---|---|---|---|---|
| | **Current situation** | **Reducing social media usage to 1 hour per day for all adolescents**[a] | **Replacing 30 min of social media usage by 30 min of physical activity**[b] | **Stopping social media usage among adolescents the most at-risk for depression**[a] |
| | Estimate (95% credible interval)/ Average percentage reduction compared to current situation | Estimate (95% credible interval)/ Average percentage reduction compared to current situation | Estimate (95% credible interval)/ Average percentage reduction compared to current situation | Estimate (95% credible interval)/ Average percentage reduction compared to current situation |
| **Cumulative prevalence of depressive symptoms onset in adolescent (millions)** | +0.59 (+0.76; +0.40)/ +26.2% | −0.33 (−0.53; −0.09)/ −14.7% | −0.29 (−0.40; −0.16)/ −12.9% | −0.27 (−0.32; −0.20)/ −12.0% |
| | Estimate (95% credible interval) | Estimate (95% credible interval) | Estimate (95% credible interval) | Estimate (95% credible interval) |
| **Cumulative lifetime prevalence of depression among adolescents (millions)** | +0.72 (+0.93; +0.50) | −0.40 (−0.65; −0.12) | −0.36 (−0.49; −0.20) | −0.33 (−0.40; −0.25) |
| **Direct medical costs (billions €)** | +2.56 (+3.29; +1.75) | −1.43 (−2.31; −0.41) | −1.26 (−1.73; −0.70) | −1.18 (−1.41; −0.87) |
| **Loss of productivity (billions €)** | +1.38 (+1.77; +0.94) | −0.77 (−1.25; −0.22) | −0.68 (−0.93; −0.38) | −0.64 (−0.76; −0.47) |
| **Total cost (billions €)** | +3.94 (+5.07; +2.70) | −2.20 (−3.56; −0.63) | −1.94 (−2.67; −1.08) | −1.81 (−2.16; −1.34) |
| **HALE loss (thousands)** | +136.87 (+175.98; +93.74) | −76.50 (−123.53; −22.04) | −67.33 (−92.69; −37.38) | −63.05 (−75.14; −46.62) |
| **Suicide attempts (thousands)** | +19.98 (+25.69; +13.68) | −11.17 (−18.03; −3.22) | −9.83 (−13.53; −5.46) | −9.20 (−10.97; −6.81) |
| **Suicide deaths** | +799 (+1,028; +547) | −447 (−721; −129) | −393 (−541; −218) | −368 (−439; −272) |

Note 1: All interventions modeled over the same study period (2000–2022) for direct comparison.

Note 2: All analyses controlled for sex, age, parental history of psychiatric disorders, childhood adversities (physical abuse, emotional abuse, sexual abuse, neglect, bullying), physical inactivity, obesity and overweight, chronic physical conditions, substance use (alcohol, tobacco, cannabis), and COVID-19 pandemic effects. Model parameters are detailed in S1 Table.

[a]Scenarios 2 and 4 (limiting social media use to 1 hour per day for all adolescents and for high-risk adolescents, respectively) were implemented over the entire 2000–2022 period.

[b]The substitution of 30 min of SM use with 30 min of physical activity applied only to adolescents with daily SM use of ≥1 h per day.

literature on absenteeism associated with depression were used [71], with average costs per day based on the French daily productivity [72]. For HALE, average duration of depression was based on prior studies [73,74], with weight based on WHO estimates [75]. For suicide and suicide attempts, we used data on lifetime suicide sex-stratified rates observed in people with depression [76], and we considered a 25:1 ratio of suicide attempts to suicide death following data from the French Observatoire National du Suicide [77].

## Interventions

All intervention scenarios were evaluated against the current situation (observed SM use and depression trends in France between 2000–2022) over the complete study period to ensure comparable assessment. We successively examined the following scenarios, for the period from 2000 to 2022: (i) The natural course of adolescent depression if no SM had been available; (ii) SM use would have been limited to a maximum of 1 hour per day. This conservative threshold was chosen as a precautionary intervention well below the 2-hour threshold identified by Sala and colleagues as associated with negative mental health outcomes in their umbrella review [78]. This threshold was selected *a priori* to capture potentially

beneficial effects of the intervention, especially on at-risk adolescents, while allowing adolescents to maintain some SM use for positive social connections and communication; and (iii) SM use would have been reduced by 30 min per day for those with a daily SM use of at least 1 hour per day and would have been replaced by 30 min per day of physical activity. Physical activity and its threshold were chosen based on evidence from a recent systematic review and network meta-analysis showing that 30 min of exercise per session has significant beneficial effects on reducing depressive symptoms in adolescents [79]. Additionally, physical activity has well-established protective effects against depression [62] and evidence suggests inverse correlations between SM use and physical activity among adolescents [30,80]. Physical activity represents a clearly modifiable behavior with direct mental health benefits; (iv) SM use would have been limited to a maximum of 1 hour per day for only the 8.5% of adolescents who are at highest risk for depression based on the known risk factors listed above.

## Model calibration

Of the 95 model parameters, 93 were obtained from prior studies, one was based on expert assumption, while one parameter, the individual baseline risk of depression, could not be estimated from the literature and was therefore calibrated (S1 Table). This baseline risk represents all unmeasured factors contributing to depression risk beyond the documented risk factors in our model. The calibration targeted the observed depression prevalence in France from l'Enquête santé européenne (EHIS) [81] and l'Enquête nationale Épidémiologie et Conditions de vie (EpiCov) (Inserm-DREES) [82] that included 85,000 people aged 15 years or more who responded to the Patient Health Questionnaire depression scale (PHQ-9) [83]. Individuals with a projected probability of depression higher than 50% were considered as having depression in the simulation model (S3 Fig). We used the Nelder-Mead simplex method [84] with convergence after 300 iterations or when the loss function fell below 0.001. The calibration process was repeated 30 times with different initial values to avoid local minima. Consistent results were observed across all 30 calibration runs, with the loss function consistently converging, suggesting robustness of the calibration process. The final calibrated baseline risk (−1.80 (SD = 1.0)) was applied to both sexes.

## Statistical analysis

The microsimulation model was run on 1,000 individuals per birth cohort from 1990 to 2012. This sample size provides adequate statistical power for the primary outcome (depression prevalence) while maintaining computational feasibility. Although this may introduce sampling variability for rare outcomes such as suicide, our Monte Carlo approach with 500 iterations based on the random variation of all non-calibrated parameters simultaneously, either within a 95%CI for parameters estimated from the literature or within a ±20% interval when CI were not available, allows us to capture this uncertainty in the credible intervals presented. This Monte Carlo approach propagated parameter uncertainty by independently sampling all 94 non-calibrated parameters in each iteration and running the model forward to observe output variability, capturing uncertainty across all model predictions. Parameters were independently sampled from predefined distributions in each simulation, distinguishing this from bootstrapping. Parameter distributions were selected based on variable nature: normal distributions for continuous variables with central tendency (e.g., baseline risk), uniform for bounded parameters without clear central values, and empirical distributions when preserved from literature. No convergence diagnostics were needed as baseline risk remained fixed. Average results and uncertainty were based on the average and distribution of the 500 Monte Carlo simulations. Results were extrapolated to the French general population exposed to SM, i.e., aged 10–17 years between 1990 and 2012, based on an average of 790,000 birth per year between 1990 and 2012.

First, we examined whether the model had adequate calibration, that is, whether it was able to adequately reproduce retrospectively (i) the observed distribution and evolution of SM use and (ii) the annual prevalence of depression in adolescents overall and by sex in the French adolescent population after 2011. Specifically, the simulated results were

respectively compared to the 4 prevalence points that were available in the literature for 2017, 2018, 2019 for SM use [52], and to the three prevalence points that were available in the literature for 2019, 2020, and 2021 [16] for adolescent depression.

Next, we examined whether the calibrated model had adequate validation. This involved evaluating whether it was able to predict prospectively the prevalence rates of adolescent depressive symptoms in another country, the United States, based on data from 167,783 adolescents aged 12–17 years who took part in the 2009–2019 waves of the NSDUH for whom past-year depression was assessed using a structured interview, a modified version of the World Health Organization Composite International Diagnostic Interview [15]. The validation model included US-specific data for physical activity [85], obesity and overweight [86], and substance use [87], calibrated the individual baseline risk to reproduce the observed depression rates in the United States in 2011 (i.e., the same year used for calibration of the model based on French data), and assumed that all other risk factors were similarly distributed in France and the United States. For both calibration and validation, model-predicted and observed parameter values were compared using $R^2$, Pearson's $R$, and visual comparison of the curves. Additionally, residual plots were generated to evaluate systematic biases and time trends patterns in prediction errors, and diagnostic plots were constructed to assess key model assumptions, including independence of residuals over time. We also performed sensitivity analyses to assess model robustness: (1) tornado diagrams examining the individual impact of each key parameter on depression prevalence estimates, (2) testing alternative standard deviation values (10% and 30%, instead of 20%) for the gamma distribution used in SM usage extrapolations, and (3) testing different correlation coefficients (0.20 and 0.40, instead of 0.10) for cross-platform SM use. Results from all sensitivity analyses are reported with their impact on the primary outcome.

Finally, we compared the outcomes of the different interventions to the current situation for the main outcome and projections for adolescent exposed to SM between 2004 and 2022.

Data were collected using Microsoft Excel. The model was performed using C++ and statistical analyses were conducted using SAS software version 9.4. The threshold for statistical significance was *a priori* fixed at two-sided $p < 0.05$. However, we prioritized effect sizes and credibility intervals over $p$-values, and presented $R^2$ and Pearson's $R$, as well as Mean Absolute Error (MAE) and Root Mean Square Error (RMSE) following current statistical practices. This study is reported as per the Transparent Reporting of a Multivariable Prediction Model for Individual Prognosis or Diagnosis (TRIPOD-AI) statement (S1 Checklist).

### Ethics statement

This study did not require ethical approval as it relied exclusively on publicly available aggregate data from national surveys and administrative databases, with no direct collection of individual participant data or human subjects involvement.

### Results

The model calibrated well based on a visually adequate fit between observed and model-predicted daily average time spent on SM (Figs 1 and S1, S2, and S4) and annual prevalence of adolescent depression in France (Fig 2), as confirmed by $R^2$ and Pearson's $R$ estimates, which were equal to 0.78 and 0.88 for SM use, and both equal to 0.97 for adolescent depression rates, respectively. Residual analysis showed no systematic bias patterns, and diagnostic plots confirmed adequate model assumptions (temporal independence) (S5 Fig). MAE and RMSE were 0.21 and 0.27 for SM use, and 0.005 and 0.006 for depression rates. Similarly, the model validated well prospectively, based on a visually adequate fit between observed and model-predicted daily average time spent on SM (Fig 1) and annual observed prevalence of adolescent depression in the USA (Fig 3), as confirmed by $R^2$ and Pearson's $R$ estimates, which were 0.90 and 0.94 for SM use and both higher than 0.97 for adolescent depression rates, respectively. MAE and RMSE for validation were 0.25 and 0.30 for SM use, and 0.013 and 0.0136 for depression rates. Model convergence is shown in S6 Fig.

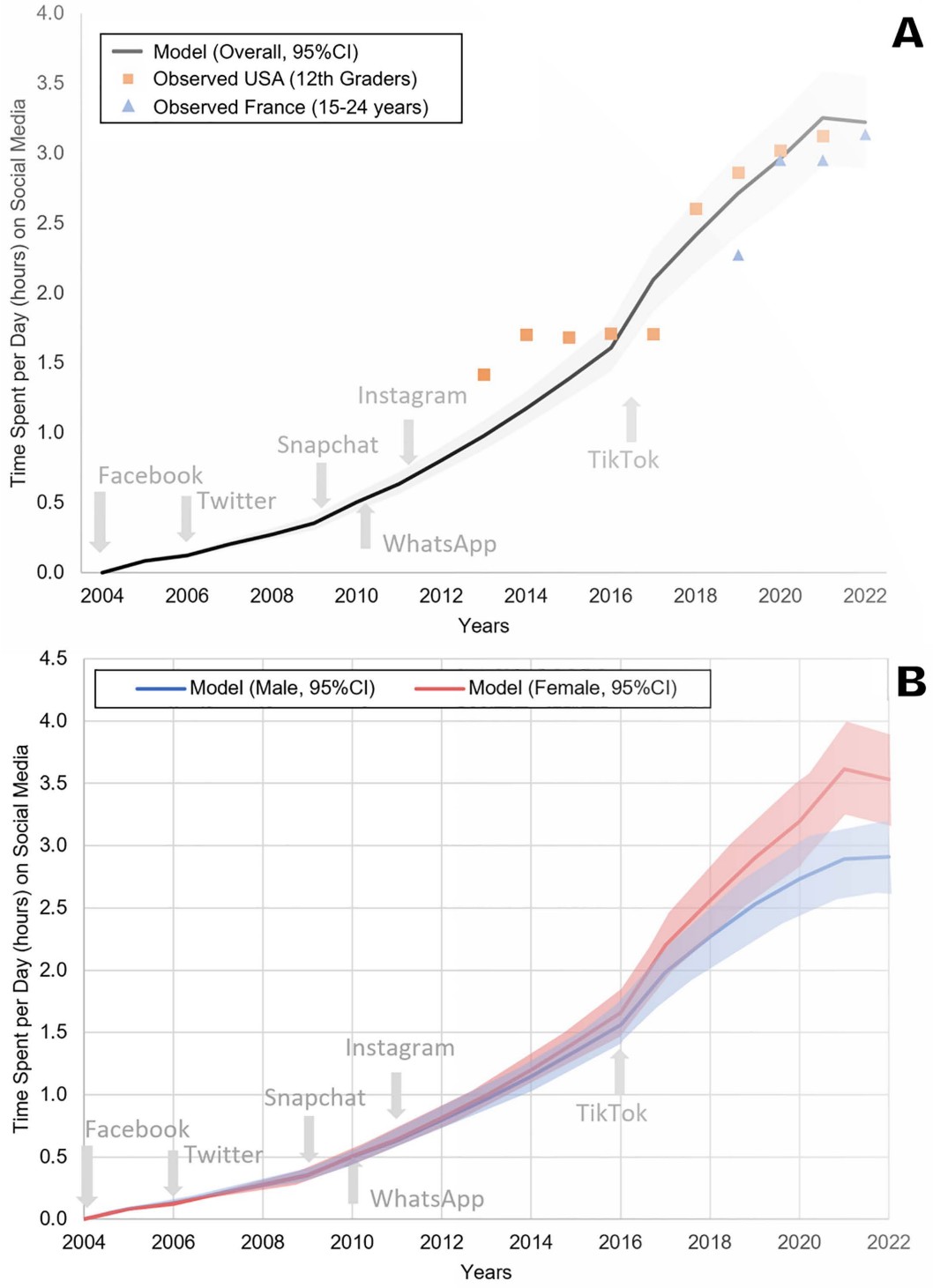

**Fig 1. Model-predicted and observed daily time spent on social media by adolescents in France and in the USA (A panel) overall, and by sex in France (B panel).** The shaded areas represent the 95% predicted uncertainty range stemming from the uncertainty in the parameter values.

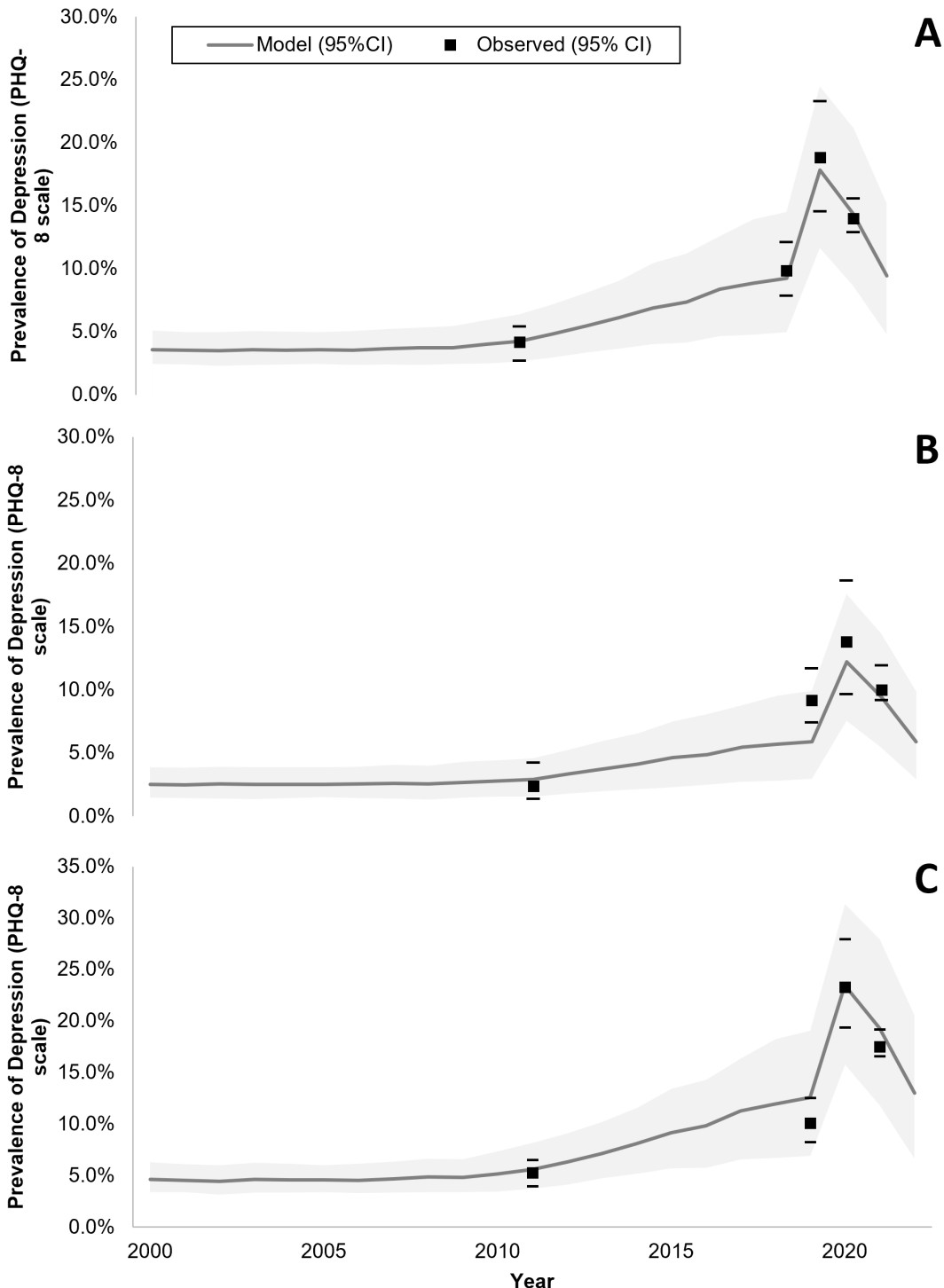

**Fig 2. Model-predicted and observed prevalence of depressive symptoms across time among female and male adolescents in France (A panel, both female and male adolescents; B panel, male adolescents; C panel, female adolescents).** The shaded areas represent the 95% predicted uncertainty range stemming from the uncertainty in the parameter values.

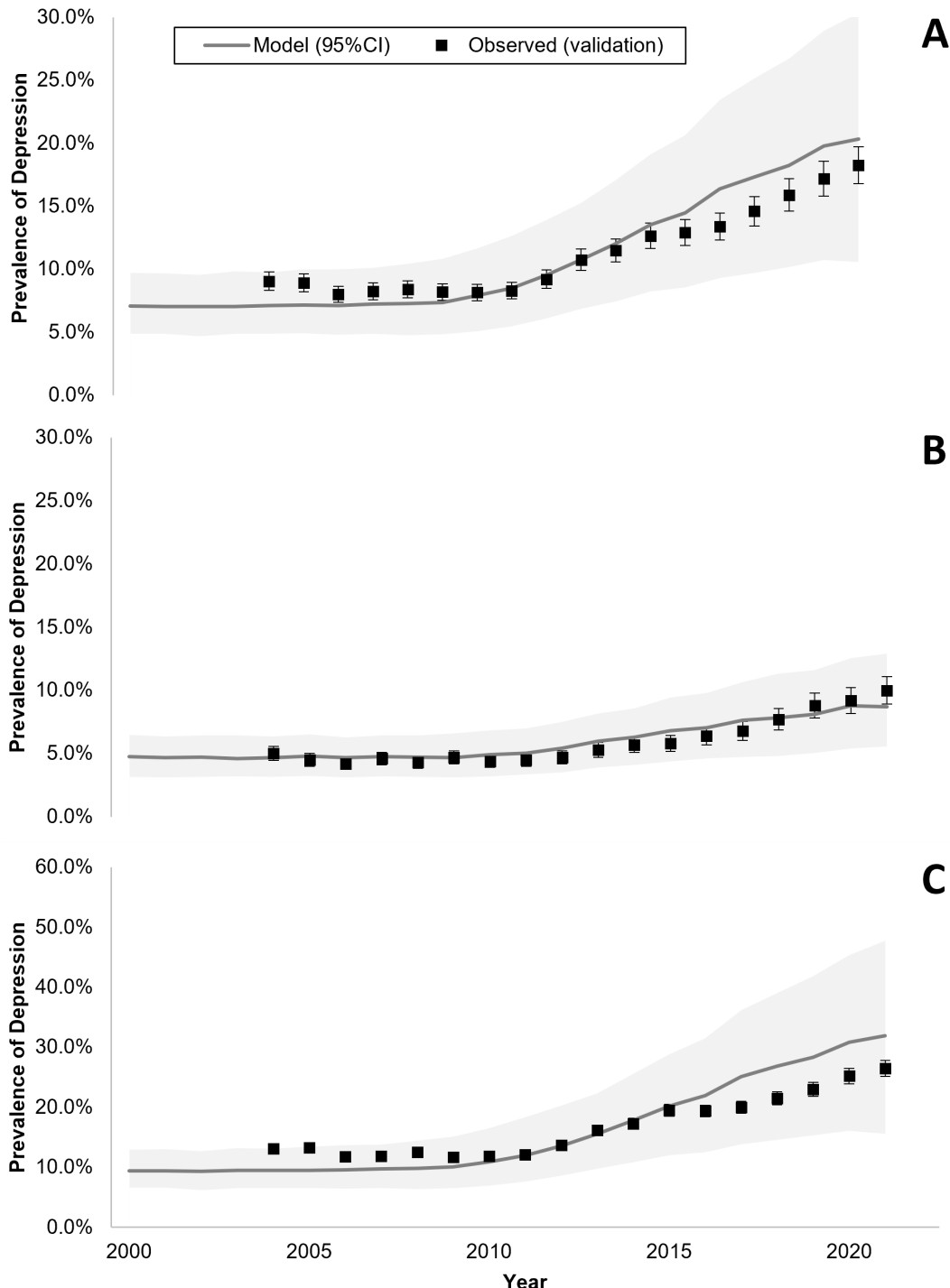

**Fig 3. Model-predicted and observed prevalence of depressive symptoms across time among female and male adolescents in the USA (A panel, both female and male adolescents; B panel, male adolescents; C panel, female adolescents).** The shaded areas represent the 95% predicted uncertainty range stemming from the uncertainty in the parameter values.

Sensitivity analyses demonstrated model robustness across all tested parameters. Tornado diagrams (S7 Fig) showed that the baseline risk parameter had the largest influence on outcomes, while SM and risk factor parameters had moderate effects. Varying the standard deviation for SM usage distributions from 10% to 30% resulted in less than 5% variation in predicted depression prevalence (S8 Fig). Similarly, changing cross-platform correlation coefficients from 0% to 30% produced less than 5% variation in predicted depression prevalence (S9 Fig).

In France, model-predicted results showed an average annual prevalence of adolescent depression of 4% in 2010 increasing to 10% in 2022 (Fig 2). Similarly, in the USA, model-predicted prevalence indicated an average annual prevalence of adolescent depression of 7% in 2004 increasing to 20% in 2021 (Fig 3). When removing the effect of SM use from the model, model-predicted prevalence remained stable in France between 2010 and 2022 (Fig 4) and in the US between 2004 and 2021 (Fig 5), with only a slight increase in adolescent depression rate in the USA, suggesting that a substantial proportion of the predicted increase in adolescent depression prevalence was related to SM use in the model.

Among the 18.6 million adolescents born in France between 1990 and 2012, we projected that existing SM use, as compared to a hypothetical scenario without SM platforms, would be associated with an additional 590 (95% credible interval (CI) [400,760]) thousands cases of depression, assuming an 82-year average life expectancy. This would translate into an additional 799 (95%CI [547,1,028] suicide deaths, 137 (95%CI [94,176]) thousands adjusted life expectancy loss (HALE) lost, and 3.94 (95%CI [2.70,5.07]) billion euros, as compared to a hypothetical scenario without SM platforms. The explained variance of depression by SM use, when considered in isolation, was low, at $R^2 = 0.04$.

On the basis of our model, we found that limiting SM use to 1 h per day for all adolescents, replacing 30 min of SM use by 30 min of physical activity, or stopping its use for the 10% of adolescents at highest risk of developing depression, would be associated with substantial benefits as compared to the current situation, as shown by a projected reduction of the cumulative prevalence of depressive symptoms by 14.7%, 12.9%, and 12.0%, respectively (Table 2). Patterns in annual adolescent depression prevalence for these different scenarios are available in Figs 4 and S10, S11, and S12. Limiting SM use to 1 hour per day for all adolescents was associated with the greatest benefits in terms of medical and social costs associated with depression across the proposed measures, generating total projected cost savings of 2.20 billion euros (95%CI [0.63, 3.56]), with all interventions modeled over the same study period (2000–2022) for direct comparison (Table 2).

## Discussion

We have proposed an individual-based microsimulation model of depression among adolescents to predict the potential impact of different measures of reduction of adolescent SM use on risk of depressive symptoms and its associated lifetime medical and societal costs in France. The model calibrated and validated well. Based on this model simulating 18.6 million adolescents born in France between 1990 and 2012, we found that excessive SM use could be a substantial independent contributor to adolescent depressive symptoms, resulting in important medical and societal costs. We projected that limiting SM use to 1 hour per day for all adolescents, replacing 30 min of SM use with 30 min of physical activity, or stopping its use for the 10% of adolescents at highest risk for depression, would substantially reduce those consequences, with the first measure potentially offering the greatest benefits.

Our study suggests that SM use is associated with depression among adolescents, independent of several well-established risk factors, including sex [56] and age, parental history of psychiatric disorders [57], exposure to childhood adversities [58–61], physical inactivity [62], obesity and overweight [39,63,64], chronic physical conditions [65], and substance use (alcohol, tobacco, and cannabis) [66]. Based on these assumptions, our results showed that simulated depression rates adequately match the observed rates in France [16] and in the USA [15]. SM use, when considered in isolation, only explained a very small proportion of depression cases ($R^2 = 0.04$), suggesting that its independent contributive role in explaining depressive states in adolescents is relatively weak. However, results also suggest that SM use may have assumed an important role in the recent increase of adolescent depression rates, especially in females. These findings

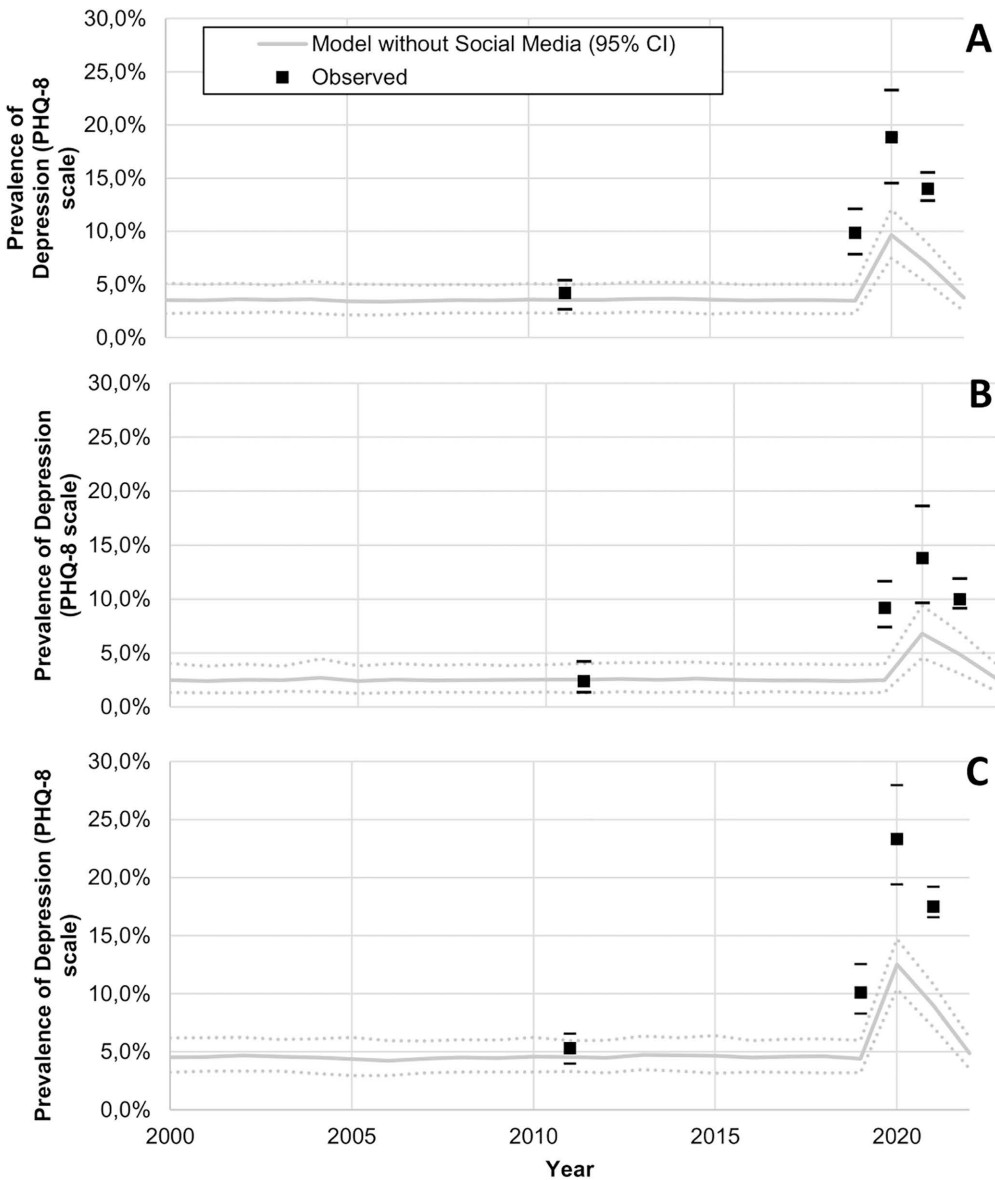

**Fig 4. Model-predicted and observed prevalence of depression across time among female and male adolescents in France for the hypothetical scenario without social media.** (**A** panel, both female and male adolescents; **B** panel, male adolescents; **C** panel, female adolescents). The solid black line represents the "business-as-usual" calibrated trajectory showing observed trends for comparison. The shaded areas represent the 95% predicted uncertainty range stemming from the uncertainty in the parameter values.

are in line with those of most prior studies [88] supporting an association of exposure to SM with depression, anxiety, and psychological distress among adolescents, although causality link in those studies was unclear due to their cross-sectional study design. While this model was calibrated to French adolescent data and validated against US adolescent data, it demonstrates potential for broader applicability with limitations. The model framework is adaptable and can be transferred to other countries with appropriate parameter adjustments (demographics, behavior patterns, risk factor distributions). However, current representativeness is limited by data availability, as many countries lack comprehensive data on adolescent SM use and adolescent depression rates. Cultural differences in SM usage and mental health contexts may also influence model applicability.

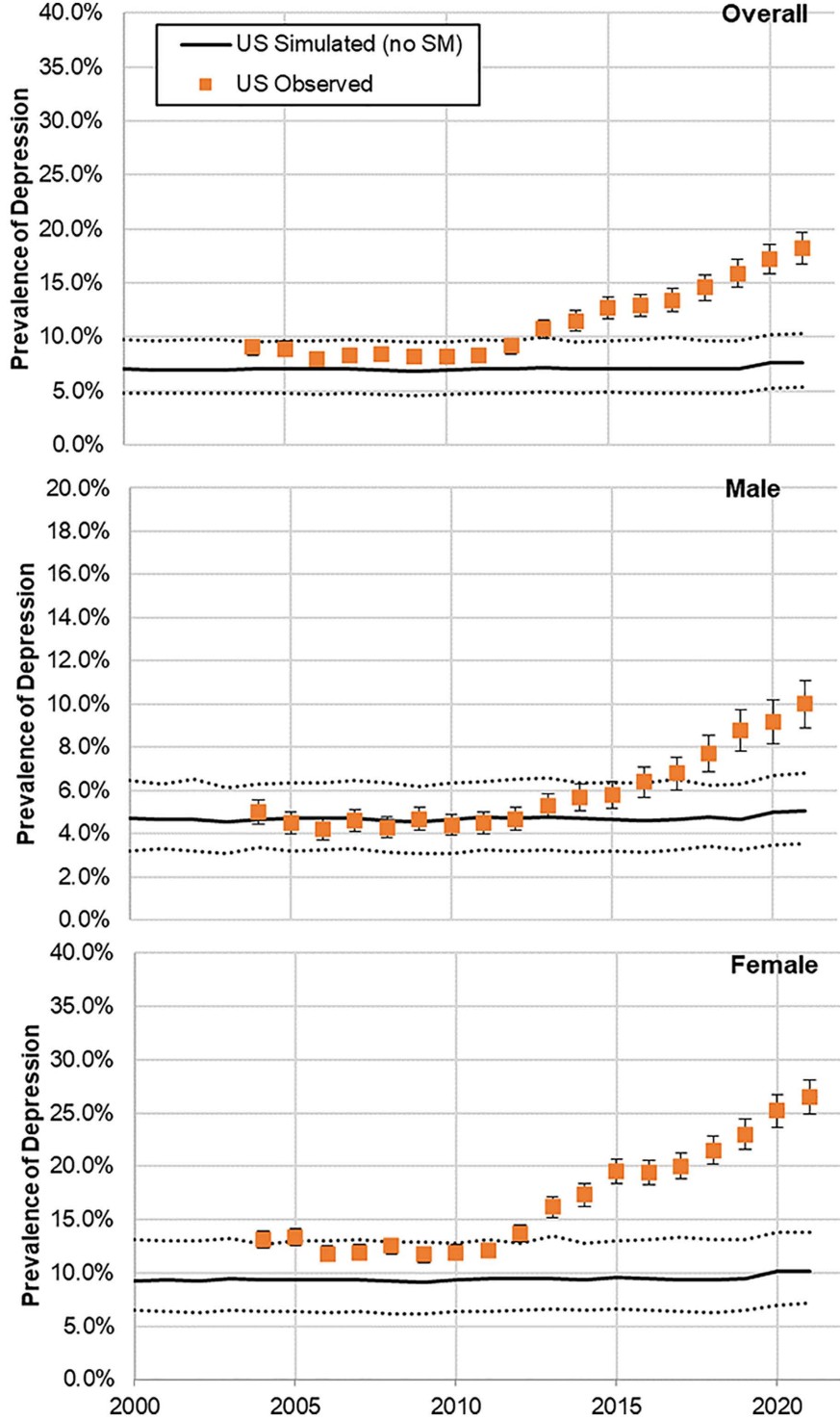

**Fig 5. Model-predicted and observed prevalence of depression across time among female and male adolescents in the USA for the hypothetical scenario without social media.** (**A** panel, both female and male adolescents; **B** panel, male adolescents; **C** panel, female adolescents). The solid black line represents the "business-as-usual" calibrated trajectory showing observed trends for comparison. The shaded areas represent the 95% predicted uncertainty range stemming from the uncertainty in the parameter values.

SM use could exacerbate preexisting psychosocial risk factors and potentially trigger depressive symptoms among adolescents through multiple potential mechanisms. First, social comparison theory suggests that constant exposure to the "Highlight Reels" on SM can lead to feelings of inadequacy and low self-esteem, which are known contributors to depression. Second, cyberbullying is prevalent on SM platforms and can result in social isolation and emotional distress [89,90]. Third, use of SM, especially before bedtime, can disrupt sleep patterns, leading to sleep deprivation and fatigue [91–94]. Fourth, the addictive nature of SM can alter motivation and brain reward circuits and lead to a decrease in real-life social interactions, which can contribute to feelings of loneliness and depressive symptoms [44,95]. Finally, the constant exposure to idealized images on SM can create unrealistic expectations and feelings of dissatisfaction [95–97].

In this microsimulation model, limiting SM use to 1 hour per day for all adolescents, replacing 30 min of SM use by 30 min of physical activity, or stopping its use for the 10% of adolescents at highest risk of developing depression, were all associated with substantially better outcomes than the current situation. Improved outcomes included lower prevalence of depression and suicide, lower adjusted life expectancy loss, and lower indirect and direct costs associated with depression. These results support that educational or public health measures contributing to even slightly reduced SM use by all adolescents (e.g., limiting use to 1 hour per day) would be expected to be beneficial, especially when considering the considerable harmful burdens of depression on the population.

Our model has several limitations. First, as with all modeling studies, we rely on existing knowledge and current assumptions. Specifically, because depressive symptoms among adolescents result from a complex interaction of social, psychological, and biological factors, our model does not likely capture all contributing factors, and its results are potentially biased due to residual unmeasured confounding (e.g., genetics). Second, our model assumes additive effects of risk factors, and does not include interaction terms between risk factors, preventing it from capturing potential synergistic effects (e.g., between childhood adversity and SM use). This additive log-linear model was chosen for parsimony and stability, as insufficient empirical evidence exists to reliably parameterize specific interactions. However, this approach is likely to have oversimplified non-linear associations between individual risk factors and depression. Third, simulating 1,000 individuals per cohort may introduce sampling variability for rare outcomes like suicide. While our uncertainty intervals account for this variability, studies focused specifically on rare outcomes might benefit from larger simulation populations. Fourth, our model did not account for seasonal variations in SM use or depression rates, potentially missing intra-annual patterns, in the absence of reliable monthly data. Fifth, depression in this report refers to depressive symptoms and not major depressive episode, as most previously published studies have focused on this potential association [32]. Therefore, our estimated of associated medical and societal costs, generally based on associations observed for major depressive episode, are possibly overestimated. Therefore, our results should not be interpreted as absolute numbers, but rather as differences in trends according to the different scenarios applied. Sixth, the COVID-19 confinement factor was modeled based on observed 2020 data, but its long-term effects on depression patterns remain uncertain and lack validation data. Seventh, although we primarily used data from national surveys with validated instruments when available, underreporting of mental health issues and inaccuracies in self-reported SM usage may have biased parameter estimates. Eight, while a strength of the microsimulation model is the opportunity to combine numerous results from different studies and account for uncertainty in parameter values, the choice of each study and estimate we included can be subject of debate. Ninth, the intervention scenarios modeled are hypothetical and not directly based on pilot studies or real-world trials. While some evidence supports individual components (e.g., physical activity benefits, SM time limits), we lack empirical data on the specific intervention designs tested here. This limits the practical applicability and real-world feasibility of our projections, as actual implementation may face unforeseen barriers, different adherence patterns, or unexpected outcomes not captured in our theoretical framework. Tenth, the intervention scenarios modeled are hypothetical and not directly based on pilot studies or real-world trials, which may limit their practical applicability and adherence rates in actual implementation. Eleventh, while we estimated in exploratory analyses the societal costs associated with depression reduction under

different scenarios, we did not perform cost-effectiveness analyses comparing intervention costs to benefits, which represents an important avenue for future research. Twelfth, our model captures SM exposure through duration of use, which represents only one dimension of a complex phenomenon. We did not model qualitative aspects such as active versus passive use, content type, or social context—factors that prior research suggests significantly moderate mental health impacts [36,41,98,99]. Future studies would benefit in incorporating these data once they will become available. Finally, while our US validation recalibrated baseline depression risk and used US-specific data for, obesity, and substance use, we assumed comparable distributions for other risk factors between countries. Regional variations in mental healthcare access, cultural attitudes toward mental health, family structures, and SM usage patterns may limit the generalizability of these assumptions. However, the adequate validation against US data suggests that the core relationships modeled may be sufficiently robust to capture depression trends across different healthcare and social systems, at least for high-income countries with similar SM penetration.

In conclusion, this microsimulation model suggests an independent association between the increase in excessive SM use by adolescents and rising rates of adolescent depression in France and the USA. Prevention measures focused on children and adolescents, especially those at-risk for depression, and their parents, should be amplified to reduce the mental health and societal burden likely linked to SM use [100]. Physicians and policymakers should take these associations under careful consideration.

## Supporting information

**S1 Fig. General shape of social media adoption curve.**
(DOCX)

**S2 Fig. Sex-stratified social media adoption curve assumption.**
(DOCX)

**S3 Fig. Projected distribution of the individual risk of depression in the simulated population using or not social media, and projected effect of using social media more than 6 h per day on this risk.**
(DOCX)

**S4 Fig. Average (95%CI) distribution of time spent on social media by adolescents simulated with the model compared to observed data in France.**
(DOCX)

**S5 Fig. Residual plots and diagnostic plots for model calibration and validation showing prediction accuracy and model assumptions.**
(DOCX)

**S6 Fig. Model convergence.**
(DOCX)

**S7 Fig. Tornado diagram showing the impact of key model parameters on depression prevalence estimates.**
(DOCX)

**S8 Fig. Sensitivity analysis testing different standard deviation values (10%–30%) for social media usage extrapolations.**
(DOCX)

**S9 Fig. Sensitivity analysis testing different correlation coefficients (0%–20%) between social media platforms.**
(DOCX)

**S10 Fig. Model-predicted and observed prevalence of depression across time among female and male adolescents in France if social media usage was limited to a maximum of 1 h per day.**
(DOCX)

**S11 Fig. Model-predicted and observed prevalence of depression across time among female and male adolescents in France, if social media usage was fully restricted for the 8.5% adolescents, the most at-risk for depression.**
(DOCX)

**S12 Fig. Model-predicted and observed prevalence of depression across time among female and male adolescents in France if 30 min of social media usage was replaced by 30 min of physical activity from 2016 onward.**
(DOCX)

**S1 Table. Summary of model parameters.**
(DOCX)

**S2 Table. Projected average time spent by adolescents on Social Media, by media.**
(DOCX)

**S1 Checklist. Transparent Reporting of a Multivariable Prediction Model for Individual Prognosis or Diagnosis (TRIPOD-AI) statement.** This checklist is reproduced from the TRIPOD Statement (BMJ 2023;378:e078378, https://doi.org/10.1136/bmj-2023-078378), and is licensed under a Creative Commons Attribution 4.0 International License (CC BY 4.0).
(PDF)

## Acknowledgments

**Disclaimer:** The views and opinions expressed in this report are those of the authors and should not be construed to represent the views of any of the sponsoring organizations, agencies, or the US government.

## Author contributions

**Conceptualization:** Nicolas Hoertel, Henri Leleu.

**Data curation:** Henri Leleu.

**Formal analysis:** Nicolas Hoertel, Henri Leleu.

**Methodology:** Nicolas Hoertel, Henri Leleu.

**Writing – original draft:** Nicolas Hoertel, Henri Leleu.

**Writing – review & editing:** Mark Olfson, Carlos Blanco, Margot Biscond, Frédéric Limosin, Marina Sánchez-Rico, Martin Blachier.

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
