## [Editor Report · Decision Letter 0]

26 Nov 2024

Dear Dr Hoertel,

Thank you for submitting your manuscript entitled "An Individual-Based Microsimulation Model of the Impact of Excessive Social Media Use On Adolescent Depression and Its Consequences in France" for consideration by PLOS Medicine.

Your manuscript has now been evaluated by the PLOS Medicine editorial staff and I am writing to let you know that we would like to send your submission out for external peer review. Please also note that we will send a rejection letter for PMEDICINE-D-24-03799.

Please re-submit your manuscript within two working days, i.e. by Nov 28 2024.

Feel free to email me at atosun@plos.org or us at plosmedicine@plos.org if you have any queries relating to your submission.

Kind regards,

Alexandra Tosun, PhD

Associate Editor

PLOS Medicine

---

## [Decision Letter · Decision Letter 1]

12 Mar 2025

Dear Dr Hoertel,

Many thanks for submitting your manuscript "An Individual-Based Microsimulation Model of the Impact of Excessive Social Media Use On Adolescent Depression and Its Consequences in France" (PMEDICINE-D-24-03990R1) to PLOS Medicine. The paper has been reviewed by subject experts and a statistician; their comments are included below and can also be accessed here: [LINK]

As you will see, the reviewers outline several points for clarification/justification and offer several suggestions for adjusting and improving the analysis. After discussing the paper with the editorial team and an academic editor with relevant expertise, I'm pleased to invite you to revise the paper in response to the reviewers' comments. We plan to send the revised paper to some or all of the original reviewers, and we cannot provide any guarantees at this stage regarding publication.

We ask that you submit your revision by Apr 02 2025. However, if this deadline is not feasible, please contact me by email, and we can discuss a suitable alternative.

Don't hesitate to contact me directly with any questions (atosun@plos.org).

Best regards,

Alexandra

Alexandra Tosun, PhD

Associate Editor

PLOS Medicine

atosun@plos.org

Comments from the academic editor:

* Please ensure that the causality in the model is sufficiently justified and discuss the evidence on which it is based, e.g. small prospective studies/trials where the key question would be the representativeness and transferability of the estimated effect on depression/outcomes of however they changed media consumption, or observational studies documenting relationships that are either cross-sectional or prospective.

* Please ensure that you have adequately adjusted for confounding and discussed potential unaddressed confounding in detail. Have you considered internet use as a coping mechanism?

* Please discuss the generalizability/transferability of the effect size.

* Please ensure that the above points are addressed and documented in detail in your point-by-point response and in the manuscript.

Comments from the reviewers:

Reviewer #1:

Major Comments:

1. The microsimulation model structure lacks transparency. Providing detailed justifications for key assumptions, particularly the calibration of residual baseline risk, would enhance the credibility of the model.

2. The model assumes additive effects of risk factors, potentially ignoring synergistic or multiplicative interactions (e.g., how childhood adversity combined with excessive social media use could amplify depression risk). Including interaction terms or testing these assumptions would improve realism.

3. The risk equation assumes a log-linear relationship between risk factors and depression, potentially oversimplifying interactions. Testing for non-linearities or complex interactions among key risk factors could improve accuracy.

4. Extrapolating social media usage with a gamma distribution assumes consistent patterns across platforms and demographic groups, which may not hold. Validation of these assumptions against independent datasets is recommended.

5. Validation against U.S. data adds robustness, but the assumption that other risk factors are similarly distributed between France and the U.S. needs stronger justification. Regional variations in mental health care access and cultural attitudes may limit generalisability.

6. The calibration process (Nelder-Mead simplex method) is standard but susceptible to local minima. Conducting sensitivity analyses to explore how variations in calibration parameters influence outcomes would strengthen the findings.

7. For bootstrapped samples, it is unclear whether convergence diagnostics were performed to ensure stability of estimates. This should be explicitly addressed.

8. Bootstrapping may underestimate uncertainty when parameter distributions are highly skewed or non-normal. Alternative methods, such as Monte Carlo simulations, should be considered to verify robustness.

9. The reliance on bootstrapping lacks evaluation of potential biases from non-independence in samples. Addressing this limitation or testing alternative resampling methods would strengthen the findings.

10. Beyond R² and Pearson's R, metrics such as mean absolute error (MAE) or root mean square error (RMSE) could provide a more nuanced assessment of model fit.

11. The model simulates annual risk assessments but does not account for intra-annual variations (e.g., seasonal effects on social media use or depression rates). Exploring these could offer more granular insights.

12. Some parameters (e.g., the 20% standard deviation for social media usage extrapolations) appear arbitrary. Providing sensitivity analyses or citing empirical evidence for these choices would improve confidence in the model.

13. The paper mentions uncertainty intervals but lacks detail on how parameter uncertainty propagates through the model. Bayesian approaches or probabilistic sensitivity analyses might better capture full uncertainty.

14. For statistical significance reporting effect sizes, confidence intervals, and advanced model fit metrics instead of p<0.05 would align better with current statistical practices.

15. While R² and Pearson's R validate the model, overfitting or predictive accuracy is not explicitly assessed. Including cross-validation or out-of-sample testing would enhance validation robustness.

16. Parameters like the 10% correlation coefficient between social media platforms seem arbitrary. Testing this assumption in sensitivity analyses would clarify its impact on results.

17. The rationale for choosing specific parameter distributions (e.g., normal or uniform) is missing. Providing this information would clarify why these distributions were used for model inputs.

Minor Comments:

18. Simulating 1,000 individuals per cohort may introduce sampling variability, especially for rare outcomes like suicide. Justification for this choice and its implications on uncertainty should be discussed.

19. While parameter values are referenced in Supplementary Table 1, a concise summary of the most critical parameters and their uncertainty ranges should be included in the main text for clarity.

20. The integration of multiple data sources (e.g., INSEE, national surveys) is commendable, but handling of inconsistencies or missing data should be explicitly detailed.

21. The "COVID-19 confinement" factor long-term effects on depression trends remain unclear. Discussing how this factor was validated or cross-checked would add robustness.

22. Scenarios like replacing 30 minutes of social media use with physical activity are idealistic. Addressing adherence and real-world feasibility would make these analyses more practical.

23. The manuscript estimates societal costs but does not address the cost-effectiveness of proposed interventions. A dedicated analysis comparing intervention costs to benefits is recommended.

24. The limitations section should explicitly address how unmeasured confounders, such as genetic predisposition, were mitigated or might bias results.

25. Hypothetical interventions are assessed but not aligned with real-world pilot studies or trials. Incorporating data from similar initiatives would validate assumptions.

26. Potential biases in data sources (e.g., underreporting of mental health issues or inaccuracies in self-reported social media usage) are not discussed and should be addressed.

27. The model assumes universal adherence to interventions, which is often unrealistic. Incorporating adherence rates from comparable interventions would make projections more realistic.

28. Figures comparing observed vs. predicted prevalence rates lack detailed annotations (e.g., reasons for discrepancies in specific years). Enhanced visual clarity would aid interpretation.

29. Figures such as Supplementary Figures 1 and 2 (adoption curves) lack explanation for assumptions about adoption trends. Clarifying whether these trends are universal or France-specific would be helpful.

30. Supplementary Table 1 could group parameters by module (e.g., demographics, behaviours, outcomes) for easier interpretation. Including metadata, such as each parameter's role (e.g., input vs. calibrated), would improve usability.

31. Supplementary Table 2 details time spent on platforms but lacks longitudinal trends. Including these trends would clarify how usage patterns influence depression risks.

32. Sensitivity results for key parameters (e.g., baseline risk, COVID-19 impact) are not detailed. Including tornado diagrams or comprehensive outputs would improve transparency.

33. While calibration is summarised (e.g., Supplementary Figure 5), the absence of residual plots or advanced goodness-of-fit metrics leaves gaps in assessing quality.

34. Figures use dotted lines for uncertainty ranges but could include shaded areas for better visual distinction.

35. Some parameter estimates (e.g., for obesity and substance use) differ slightly between tables and figures. Consistency or explanations for discrepancies would avoid confusion.

Reviewer #2:

This paper developed an individual-based microsimulation to model the impact of excessive social media use on depression among adolescents in France. The model was calibrated to reproduce observed rates of depression in France in 2011. The calibrated model was further tested to see whether it reproduced empirical trends in social media use prevalence and observed rates of depression in France after 2011. Model validation was assessed to determine the model's ability to predict the prevalence of adolescent depressive symptoms in the United States using 2009-2019 data. Finally, the validated model was used to explore four scenarios and their impacts on total costs and mortality attributed to depression and suicide:

1. The natural course of adolescent depression (if social media had not been available over the period 2000-2022);

2. A scenario where social media use would have been limited to a maximum of 1 hour per day;

3. Starting in 2016, a scenario where social media was reduced by 30 minutes per day for those with daily use ≥ 1 hour per day and that time replaced with 30 minutes of physical activity per day;

4. Social media limited to a maximum of 1 hour per day for 8.5% of adolescents at highest risk for depression.

The authors should be commended for their work in developing such a novel model and their rigorous approach to model calibration and validation to ensure the model produced meaningful results on a timely and important issue.

Below are a few comments and suggestions intended to further strengthen key aspects of the paper.

Introduction

- The authors provide a well-reasoned argument for the focus on social media and the challenges in disentangling its influence on rising adolescent depression rates. What is missing however is a clear explication of the gap this model is trying to fill. Why was individual-based microsimulation modelling used over other approaches e.g., agent-based modelling? Have studies previously used this approach to understand social media use (to my knowledge, they have not which highlights the novelty of this model)?

- It is important also to provide some context for the choice of intervention scenarios being modelled. For example, one of the scenarios relates to time substitution from social media use to physical activity. Why was physical activity chosen? Is there any evidence to suggest that adolescents who engage in excessive social media use are less physically active? Is it not equally possible that time might be substituted from other activities e.g., sleeping which is also a risk factor for depressive symptoms? To be clear, the choice to look at physical activity is not an issue, I just think the paper would be strengthened by providing a better context around time substitution and evidence supporting the choice to look at physical activity over other activities.

Methods

- Could authors clarify if DALE - adjusted life expectancy loss - refers to Disability-adjusted life years (DALYs)? Or if authors are referring to Health-adjusted life expectancy (HALE)? DALE may confuse some readers as its not a term I've come across previously.

- The chosen interventions are interesting however, their effects are difficult to compare as they are all implemented over different time periods. Within the figures, it would be helpful to provide a line that shows the 'business-as-usual' or calibrated trajectory which shows the trends observed in the data, enabling readers to make a visual comparison of the potential impact of each scenario relative to the baseline case.

- What was the timeframe for scenario 2 where social media use is limited to a maximum of 1 hour per day and scenario 4 where these limits were only imposed on high-risk adolescents? Were these over the same time period 2000-2022?

- Why did authors decide to limit the timeframe of scenario 3 to 2016 and onwards? It makes it hard to compare which scenario results in the greatest reduction in the cumulative prevalence of depression when the timeframes are so different. This scenario seems almost as effective as scenario 2 where social media is limited to 1 hour max for all and makes me wonder whether it would have been more effective if it were implemented from 2000-2022 and not 2016-2022.

- A recent umbrella review by Sala focusing on the mental health impacts of social media use among adolescents found that > 2 hours of social media use per day was associated with negative mental health outcomes. It would also be helpful if authors could provide some justification or insight into why a 1 hour cut off for social media us was tested and others weren't considered.

Sala, A., L. Porcaro, and E. Gómez, Social Media Use and adolescents' mental health and well-being: An umbrella review. Computers in Human Behavior Reports, 2024. 14: p. 100404.

Results

- When authors mention 'the current situation' are they referring to the observed social media and depression trends or the scena

- On page 12, at the bottom of the results section authors state that "limiting social media use to 1 hour per day for all adolescents was associated [with] the greatest benefits in terms of medical and social costs…". This makes sense in relation to Table 1 which states that interventions all spanned 2000-2022. But per my earlier point, this seems to conflict with the methods section (p. 8) which states that "Starting in 2016 when average exposure to SM was greater than 90 min, SM use would have been reduced by 30 min per day …". Can authors please clarify the timeframes in both the text and table?

- If one intervention was only implemented from 2016 onwards, and the other from 2000, then I think it would be misleading to say that one is more effective than the other without explicitly qualifying this statement and the extent to which these different timeframes likely impacted the findings.

- Table 1, could authors clarify that scenario 2 where 30 minutes of social media use is replaced by 30 min of physical activity, that this replacement only occurs 'for those with daily use ≥ 1 hour per day.

Discussion

- Could authors make a statement about the generalisability of the model's findings and the adaptability of the model to other contexts?

Figure S2

- Could authors explain in more detail how the curves in Appendix figure S2 were derived and the assumptions underpinning these?

Minor comments

- On page 9, under section 2.7, "…general population exposed to SN" and page 10, "…adolescents exposed to SN between…" - please change to SM. More generally please define social media as SM and use the acronym consistently in all subsequent mentions.

- On page 12, at the bottom of the results section authors state that "limiting social media use to 1 hour per day for all adolescents was associated [with] the greatest benefits in terms of medical and social costs…". I think there is a 'with' missing.

Reviewer #3: Thank you for submitting this very informative and well written manuscript titled An Individual-Based Microsimulation Model of the Impact of Excessive Social Media Use On Adolescent Depression and Its Consequences in France.

Overall, this is an excellent piece and an important topic to consider.

I believe the background literature and rationale are mostly sufficient. However, I believe there are some more recent studies that could be used in some sections to enhance the robustness of the assumptions. There are some very recent papers outlining links (or lack of evidence of links) between SM use and psychological distress. The recent literature suggests that it is not as simple as more time on SM = increases in mental health problems. The relationship is complex and affected by several factors such as engagement with SM (i.e. active versus passive engagement), experiences with cyberbullying, who and how young people are interacting with (i.e. friend or foe) and their own engagement in terms of posting. This needs to be accounted for or considered in more detail at least in the intro and discussion, and perhaps in the model if at all possible.

In section 2 (Methods) I would like to see a few sentences to summarize the method rather than just referring to previous studies. Currently it says "Following previously described methods [27-31]".

The model description and justification are clear and detailed, and the purpose and scope are evident and justified.

It appears that the data used in the model are appropriate to the context and population under study, and have been drawn from recent statistics or (mostly) recent studies - based on my earlier comment there may perhaps be some that could be more updated.

Parameters seem to be appropriately selected.

Validation and calibration are well reported.

Can the authors expand on any sensitivity analysis carried out in terms of if any assumptions or parameters were changed did this affect the model outcomes?

Results and interpretation are good. Perhaps with the addition of further literature as suggested above, this may need changing though.

Model documentation is clear and appears reproducible.

---

* Please upload any figures associated with your paper as individual TIF or EPS files with 300dpi resolution at resubmission; please read our figure guidelines for more information on our requirements: http://journals.plos.org/plosmedicine/s/figures. While revising your submission, please upload your figure files to the PACE digital diagnostic tool, https://pacev2.apexcovantage.com/. PACE helps ensure that figures meet PLOS requirements. To use PACE, you must first register as a user. Then, login and navigate to the UPLOAD tab, where you will find detailed instructions on how to use the tool. If you encounter any issues or have any questions when using PACE, please email us at PLOSMedicine@plos.org.

* FINANCIAL DISCLOSURES: The funding statement should include: specific grant numbers, initials of authors who received each award, URLs to sponsors’ websites. Also, please state whether any sponsors or funders (other than the named authors) played any role in study design, data collection and analysis, the decision to publish, or preparation of the manuscript. If they had no role in the research, include this sentence: “The funders had no role in study design, data collection and analysis, decision to publish, or preparation of the manuscript.”

* COMPETING INTEREST: All authors must declare their relevant competing interests per the PLOS policy, which can be seen here: https://journals.plos.org/plosmedicine/s/competing-interests

For authors with ties to industry, please indicate whether any of the interests has a financial stake in the results of the current study.

* DATA AVAILABILITY: The Data Availability Statement (DAS) requires revision. For each data source used in your study:

* ETHICS STATEMENT: Please include a short statement on why your study did not require ethical approval.

FIGURES AND TABLES

SUPPLEMENTARY MATERIAL

REFERENCES

STUDY TYPE-SPECIFIC REQUESTS

The following list is derived from Geoffrey P Garnett, Simon Cousens, Timothy B Hallett, Richard Steketee, Neff Walker. Mathematical models in the evaluation of health programmes. (2011) Lancet DOI:10.1016/S0140-6736(10)61505-X:

* If pertinent, please provide a diagram that shows the model structure, including how the natural history of the disease is represented, the process and determinants of disease acquisition, and how the putative intervention could affect the system.

* Please provide a complete list of model parameters, including clear and precise descriptions of the meaning of each parameter, together with the values or ranges for each, with justification or the primary source cited and important caveats about the use of these values noted.

* Please provide a clear statement about how the model was fitted to the data, including goodness-of-fit measure, the numerical algorithm used, which parameter varied, constraints imposed on parameter values, and starting conditions.

* For uncertainty analyses, please state the sources of uncertainties quantified and not quantified [can include parameter, data, and model structure].

* Please provide sensitivity analyses to identify which parameter values are most important in the model. Uncertainty estimates seek to derive a range of credible results on the basis of an exploration of the range of reasonable parameter values. The choice of method should be presented and justified.

* Please discuss the scientific rationale for the choice of model structure and identify points where this choice could influence conclusions drawn. Please also describe the strength of the scientific basis underlying the key model assumptions.

* For studies that develop a prediction model or evaluate its performance, please ensure that the study is reported according to the TRIPOD statement (https://www.equator-network.org/reporting-guidelines/tripod-statement) and include the completed checklist as Supporting Information. Please add the following statement, or similar, to the Methods: "This study is reported as per the Transparent Reporting of a Multivariable Prediction Model for Individual Prognosis Or Diagnosis (TRIPOD) statement (S1 Checklist)." For studies using machine learning, please use the TRIPOD-AI checklist. When completing the checklist, please use section and paragraph numbers, rather than page numbers.

---

## [Decision Letter · Decision Letter 2]

6 Aug 2025

Dear Dr. Hoertel,

Thank you very much for re-submitting your manuscript "An Individual-Based Microsimulation Model of the Impact of Excessive Social Media Use On Adolescent Depression and Its Consequences in France" (PMEDICINE-D-24-03990R2) for review by PLOS Medicine.

Thank you for your detailed response to the reviewers' and editors’ comments. I have discussed the paper with my colleagues, and it has also been seen again by all of the original reviewers. The changes made to the paper were satisfactory to the reviewers. As such, we intend to accept the paper for publication, pending your attention to the editors' comments below in a further revision. When submitting your revised paper, please once again include a detailed point-by-point response to the editorial comments.

The remaining issues that need to be addressed are listed at the end of this email. Any accompanying reviewer attachments can be seen via the link below. Please take these into account before resubmitting your manuscript: ********

In revising the manuscript for further consideration here, please ensure you address the specific points made by the editors. In your rebuttal letter you should indicate your response to the editors' comments and the changes you have made in the manuscript. Please submit a clean version of the paper as the main article file. A version with changes marked must also be uploaded as a marked up manuscript file. Please also check the guidelines for revised papers at http://journals.plos.org/plosmedicine/s/revising-your-manuscript for any that apply to your paper.

We ask that you submit your revision within 1 week (Aug 13 2025). However, if this deadline is not feasible, please contact me by email, and we can discuss a suitable alternative.

Please do not hesitate to contact me directly with any questions (atosun@plos.org).

We look forward to receiving the revised manuscript.

Sincerely,

Alexandra Tosun, PhD

Senior Editor 

PLOS Medicine

plosmedicine.org

Comments from Reviewers:

Reviewer #1: Thank you for thoroughly addressing all my comments and providing detailed clarifications in your revised manuscript. I appreciate the effort you have put into enhancing the rigor and transparency of your analyses.

The manuscript now reflects a robust and valuable contribution to subject, I am happy to recommend this work for publication and look forward to seeing its positive impact in the field.

Reviewer #2: The authors have done a great job revising the manuscript and have thoroughly addressed all the points from my review. This is a strong piece of work that will make a valuable contribution to the field. Congratulations to the authors.

Reviewer #3: Thank you to the authors for thoroughly attending to all the reviewer comments in such detail. This has further improved the quality and transparency of the study. Well done.

********

Requests from Editors:

* Please confirm that your title complies with to PLOS Medicine's style. Your title must be nondeclarative and not a question. It should begin with main concept if possible. "Effect of" should be used only if causality can be inferred, i.e., for an RCT. Please place the study design ("A randomized controlled trial," "A retrospective study," "A modelling study," etc.) in the subtitle (i.e., after a colon).

* Statistical reporting: Please revise throughout the manuscript, including tables and figures.

- Please report statistical information as follows to improve clarity for the reader ""22% (95% CI [13,28]; p</=)"".

- Please separate upper and lower bounds with commas instead of hyphens as the latter can be confused with reporting of negative values.

- Please repeat statistical definitions (HR, CI etc.) for each set of parentheses.

* Please ensure that all abbreviations are defined at first use throughout the text (including statistical abbreviations).

* Please ensure that tables and figures, including those in supplementary files, are appropriately referenced in the main text.

* Please review your text for claims of novelty or primacy (e.g. 'for the first time' or ‘novel’) and remove this language.

* Please check that any use of statistical terms (such as trend or significant) are supported by the data, and if not please remove them. The term trend should be used only when the test for trend has been conducted.

* Please provide titles and legends for all figures and tables (including those in Supporting Information files). Please define all acronyms used in each figure or table in its corresponding legend.

* Please remove the numbering from the headings.

* Please confirm that the information you provided in the online submission form is up to date and accurate.

ABSTRACT

* Please confirm that your abstract complies with our requirements, including providing all the information relevant to this study type https://journals.plos.org/plosmedicine/s/submission-guidelines#loc-abstract

* Abstract Background: The final sentence should clearly state the study question.

* Please include the study design, population and setting, number of participants, years during which the study took place, length of follow up, and main outcome measures.

* Please quantify the main results (with 95% CIs and p values).

* Please include the important dependent variables that are adjusted for in the analyses (if too many, please sure overarching categories).

* Please ensure that all numbers presented in the abstract are present and identical to numbers presented in the main manuscript text.

* Abstract Conclusions: Please address the study implications without overreaching what can be concluded from the data; the phrase "In this study, we observed ..." may be useful. Please interpret the study based on the results presented in the abstract, emphasizing what is new without overstating your conclusions.

AUTHOR SUMMARY

* Under ‘What Did the Researchers Do and Find?’, we suggest adding an additional bullet point that briefly explains what the aim of the microsimulation model was.

* In the final bullet point of 'What Do These Findings Mean?', please include the main limitations of the study in non-technical language.

INTRODUCTION

* Please ensure that the Introduction ends with a clear description of the study question or hypothesis.

METHODS AND RESULTS

* Please add the following statement, or similar, to the Methods: "This study is reported as per the Transparent Reporting of a Multivariable Prediction Model for Individual Prognosis Or Diagnosis (TRIPOD-AI) statement (S1 Checklist)." When completing the checklist, please use section and paragraph numbers, rather than page numbers.

* When reporting age, please add a unit, such as years.

* l.292: Please note that this information (Data availability and Code availability) should be included in the Data Availability Statement in the online submission form. You may remove the links from the main text and simply describe the data sources and how they were used.

* Figure 1: Please define the meaning of the orange triangles in the figure legend. Please note that the figure description appears to repeat certain information. Please revise.

* l.356ff, “Limiting social media use to 1 hour per day for all adolescents was associated with the greatest benefits in terms of medical and social costs…” – we suggest describing the cost results in more detail here.

* Please specify the variables controlled for in all relevant Tables.

* We find the findings in Figures S10 and S11 very interesting and useful. They might be worth moving from the supplementary information to the main text. However, we will leave this decision to you.

General Editorial Requests

---

## [Editor Report · Decision Letter 3]

29 Aug 2025

Dear Dr Hoertel, 

On behalf of my colleagues and the Academic Editor, Jeremy D Goldhaber-Fiebert, I am pleased to inform you that we have agreed to publish your manuscript "Impact of Excessive Social Media Use On Adolescent Depression and Its Consequences in France: An Individual-Based Microsimulation Model" (PMEDICINE-D-24-03990R3) in PLOS Medicine.

I appreciate your thorough responses to the reviewers' and editors' comments throughout the editorial process. We look forward to publishing your manuscript. Editorially, there are a few remaining points that should be addressed prior to publication. We will carefully check whether the changes have been made. If you have any questions or concerns regarding these final requests, please feel free to contact me at atosun@plos.org.

Please see below the minor points that we request you respond to:

1) Data availability: In the Data Availability Statement of the online submission form, please include a statement indicating that all underlying data are publicly available and described in the manuscript with relevant references.

2) Data availability: Please confirm that the data on prevalence rates of adolescent depression in France and in the United States are also publicly available. If so, please clearly state so.

3) Code availability: Because Github depositions can be readily changed or deleted, we encourage you to make a permanent DOI'd copy (e.g. in Zenodo) and provide the URL.

4) Figure 5: Please add an x-axis label.

5) Please note that the headings (and subheadings) are still numbered. Please remove.

Before your manuscript can be formally accepted you will need to complete some formatting changes, which you will receive in a follow up email (including the editorial requests above). Please be aware that it may take several days for you to receive this email; during this time no action is required by you. Once you have received these formatting requests, please note that your manuscript will not be scheduled for publication until you have made the required changes.

PRESS

Sincerely, 

Alexandra Tosun, PhD 

Senior Editor 

PLOS Medicine